# RECONSTRUCTING CONTINUOUS DISTRIBUTIONS OF 3D PROTEIN STRUCTURE FROM CRYO-EM IMAGES

**Ellen D. Zhong**
MIT
zhonge@mit.edu

**Tristan Bepler**
MIT
tbepler@mit.edu

**Joseph H. Davis**[*]
MIT
jhdavis@mit.edu

**Bonnie Berger**[*]
MIT
bab@mit.edu

## ABSTRACT

Cryo-electron microscopy (cryo-EM) is a powerful technique for determining the structure of proteins and other macromolecular complexes at near-atomic resolution. In single particle cryo-EM, the central problem is to reconstruct the 3D structure of a macromolecule from $10^{4-7}$ noisy and randomly oriented 2D projection images. However, the imaged protein complexes may exhibit structural variability, which complicates reconstruction and is typically addressed using discrete clustering approaches that fail to capture the full range of protein dynamics. Here, we introduce a novel method for cryo-EM reconstruction that extends naturally to modeling continuous generative factors of structural heterogeneity. This method encodes structures in Fourier space using coordinate-based deep neural networks, and trains these networks from unlabeled 2D cryo-EM images by combining exact inference over image orientation with variational inference for structural heterogeneity. We demonstrate that the proposed method, termed cryoDRGN, can perform *ab initio* reconstruction of 3D protein complexes from simulated and real 2D cryo-EM image data. To our knowledge, cryoDRGN is the first neural network-based approach for cryo-EM reconstruction and the first end-to-end method for directly reconstructing continuous ensembles of protein structures from cryo-EM images.

## 1 INTRODUCTION

Cryo-electron microscopy (cryo-EM) is a Nobel Prize-winning technique capable of determining the structure of proteins and macromolecular complexes at near-atomic resolution. In a single particle cryo-EM experiment, a purified solution of the target protein or biomolecular complex is frozen in a thin layer of vitreous ice and imaged at sub-nanometer resolution using an electron microscope. After initial preprocessing and segmentation of the raw data, the dataset typically comprises $10^{4-7}$ noisy projection images. Each image contains a separate instance of the molecule, recorded as the molecule's electron density integrated along the imaging axis (Figure 1). A major bottleneck in cryo-EM structure determination is the computational task of 3D reconstruction, where the goal is to solve the inverse problem of learning the structure, i.e. the 3D electron density volume, which gave rise to the projection images. Unlike classic tomographic reconstruction (e.g. MRI), cryo-EM reconstruction is complicated by the unknown orientation of each copy of the molecule in the ice. Furthermore, cryo-EM reconstruction algorithms must handle challenges such as an extremely low signal to noise ratio (SNR), unknown in-plane translations, imperfect signal transfer due to microscope optics, and discretization of the measurements. Despite these challenges, continuing advances in hardware and software have enabled structure determination at near-atomic resolution for *rigid* proteins (Kühlbrandt (2014); Scheres (2012b); Renaud et al. (2018); Li et al. (2013)).

Many proteins and other biomolecules are intrinsically flexible and undergo large conformational changes to perform their function. Since each cryo-EM image contains a unique instance of the molecule of interest, cryo-EM has the potential to resolve structural heterogeneity, which is experimentally infeasible with other structural biology techniques such as X-ray crystallography. However, this heterogeneity poses a substantial challenge for reconstruction as each image is no longer of the same structure. Traditional reconstruction algorithms address heterogeneity with discrete clustering

---

[*]Corresponding authors

approaches, however, protein conformations are continuous and may be poorly approximated with discrete clusters (Malhotra & Udgaonkar (2016); Nakane et al. (2018)).

Here, we introduce a neural network-based reconstruction algorithm that learns a continuous low-dimensional manifold over a protein's conformational states from unlabeled 2D cryo-EM images. We present an end-to-end learning framework for a generative model over 3D volumes using an image encoder-volume decoder neural network architecture. Extending spatial-VAE, we formulate our decoder as a function of 3D Cartesian coordinates and unconstrained latent variables representing factors of image variation that we expect to result from protein structural heterogeneity (Bepler et al. (2019)). All inference is performed in Fourier space, which allows us to efficiently relate 2D projections to 3D volumes via the Fourier slice theorem. By

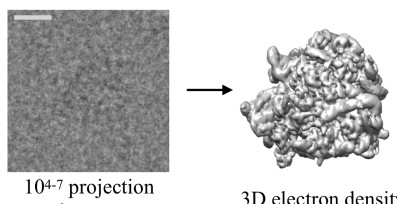

10^{4-7} projection images     3D electron density

Figure 1: Cryo-EM reconstruction algorithms tackle the inverse problem of determining the 3D electron density volume from $10^{4-7}$ noisy images. Each image is a noisy projection of a unique instance of the molecule suspended in ice at a random orientation. Algorithms must jointly learn the volume and the orientation of each particle image. Example image from Wong et al. (2014).

formulating our decoder as a function of Cartesian coordinates, we can explicitly model the imaging operation to disentangle the orientation of the molecule during imaging from intrinsic protein structural heterogeneity. Our learning framework avoids errant local minima in image orientation by optimizing with exact inference over a discretization of $SO(3) \times \mathbb{R}^2$ using a branch and bound algorithm. The unconstrained latent variables are trained in the standard variational autoencoder approach. We present results on both real and simulated cryo-EM data.

## 2 BACKGROUND AND NOTATION

### 2.1 IMAGE FORMATION MODEL

Cryo-EM aims to recover a structure of interest $V : \mathbb{R}^3 \to \mathbb{R}$ consisting of an electron density at each point in space based on a collection of noisy images $X_1, ..., X_N$ produced by projecting (i.e. integrating) the volume in an unknown orientation along the imaging axis. Formally, the generation of image $X$ can be modeled as:

$$X(r_x, r_y) = g * \int_{\mathbb{R}} V(R^T \mathbf{r} + t) \, dr_z + noise \qquad \mathbf{r} = (r_x, r_y, r_z)^T \qquad (1)$$

where $V$ is the electron density (*volume*), $R \in SO(3)$, the 3D rotation group, is an unknown orientation of the volume, and $t = (tx, ty, 0)$ is an unknown in-plane translation, corresponding to imperfect centering of the volume within the image. The image signal is convolved with $g$, the point spread function for the microscope before being corrupted with frequency-dependent noise and registered on a discrete grid of size DxD, where D is the size of the image along one dimension.

The reconstruction problem is simplified by the observation that the Fourier transform of a 2D projection of $V$ is a 2D slice through the origin of $V$ in the Fourier domain, where the slice is perpendicular to the projection direction. This correspondence is known as the *Fourier slice theorem* (Bracewell (1956)). In the Fourier domain, the generative process for image $\hat{X}$ from volume $\hat{V}$ can thus be written:

$$\hat{X}(k_x, k_y) = \hat{g}S(t)A(R)\hat{V}(k_x, k_y) + \epsilon \qquad (2)$$

where $\hat{g} = \mathcal{F}g$ is the contrast transfer function (CTF) of the microscope, $S(t)$ is a phase shift operator corresponding to image translation by $t$ in real space, and $A(R)\hat{V} = \hat{V}(R^T(\cdot, \cdot, 0)^T)$ is a linear slice operator corresponding to rotation by $R$ and linear projection along the z-axis in real space. The frequency-dependent noise $\epsilon$ is typically modelled as independent, zero-centered Gaussian noise in Fourier space. Under this model, the probability of of observing an image $\hat{X}$ with pose $\phi = (R, t)$ from volume $\hat{V}$ is thus:

$$p(\hat{X}|\phi, \hat{V}) = p(\hat{X}|R, t, \hat{V}) = \frac{1}{Z} \exp \left( \sum_l \frac{-1}{2\sigma_l^2} \left| \hat{g}_l A_l(R)\hat{V} - S_l(t)\hat{X}_l \right|^2 \right) \qquad (3)$$

where $l$ is a two-component index over Fourier coefficients for the image, $\sigma_l$ is the width of the Gaussian noise expected at each frequency, and $Z$ is a normalization constant.

## 2.2 TRADITIONAL CRYO-EM RECONSTRUCTION

To recover the desired structure, cryo-EM reconstruction methods must jointly solve for the unknown volume $V$ and image poses $\phi_i = (R_i, t_i)$. Expectation maximization (Scheres (2012a)) and simpler variants of coordinate ascent are typically employed to find a *maximum a posteriori* estimate of $V$ marginalizing over the posterior distribution of $\phi_i$'s, i.e.:

$$V^{\text{MAP}} = \arg\max_V \sum_{i=1}^{N} \log \int p(X_i|\phi, V)p(\phi)d\phi + \log p(V) \qquad (4)$$

Intuitively, given $V^{(n)}$, the estimate of the volume at iteration $n$, images are first aligned with $V^{(n)}$ (E-step), then with the updated alignments, the images are backprojected to yield $V^{(n+1)}$ (M-step). This iterative refinement procedure is sensitive to the initial estimate of $V$ as the optimization objective is highly nonconvex; stochastic gradient descent is commonly used for *ab initio* reconstruction[1] to provide an initial estimate $V^{(0)}$ (Punjani et al. (2017)).

Given sample heterogeneity, the standard approach in the cryo-EM field is to simultaneously reconstruct $K$ independent volumes. Termed *multiclass refinement*, the image formation model is extended to assume images are generated from $V_1, ..., V_K$ independent volumes, with inference now requiring marginalization over $\phi_i$'s and class assignment probabilities $\pi_j$'s:

$$\arg\max_{V_1, ..., V_K} \sum_{i=1}^{N} \log \sum_{j=1}^{K} \left( \pi_j \int p(X_i|\phi, V_j)p(\phi)d\phi \right) + \sum_{j=1}^{K} \log p(V_j) \qquad (5)$$

While this formulation is sufficiently descriptive when the structural heterogeneity consists of a small number of discrete conformations, it suffers when the heterogeneity is complex or when conformations lie along a continuum of states. In practice, resolving such heterogeneity is handled through a hierarchical approach refining subsets of the imaging dataset with manual choices for the number of classes and the initial models for refinement. Because the number and nature of the underlying structural states are unknown, multiclass refinement is error-prone, and in general, the identification and analysis of heterogeneity is an open problem in single particle cryo-EM.

## 3 METHODS

We propose a neural network-based reconstruction method, cryoDRGN (Deep Reconstructing Generative Networks), that can perform *ab-initio* unsupervised reconstruction of a continuous distribution over 3D volumes from unlabeled 2D images (Figure 2). We formulate an image encoder-volume decoder architecture based on the variational autoencoder (VAE) (Kingma & Welling (2013)), where protein structural heterogeneity is modeled in the latent variable. While a standard VAE assumes all sources of image heterogeneity are entangled in the latent variable, we propose an architecture that enables modelling the intrinsic heterogeneity of the volume separately from the extrinsic orientation of the volume during imaging. Our end-to-end training framework explicitly models the forward image formation process to relate 2D views to 3D volumes and employs two separate strategies for inference: a variational approach for the unconstrained latent variables and a global search over $SO(3) \times \mathbb{R}^2$ for the unknown pose of each image. These elements are described in further detail below.

---

[1]"Reconstruction" is used interchangeably in the cryo-EM literature to refer to either the full pipeline from *ab-initio* model generation followed by iterative refinement of the model via expectation maximization or solely to the latter. We focus on the former case.

### 3.1 GENERATIVE MODEL

We design a deep generative model to approximate a single function, $\hat{V} : \mathbb{R}^{3+n} \to \mathbb{R}$, representing a n-dimensional manifold of 3D electron densities in the Fourier domain. Specifically, the volume $\hat{V}$ is modelled as a probabilistic decoder $p_\theta(\hat{V}|k, z)$, where $\theta$ are parameters of a multilayer perceptron (MLP). Given Cartesian coordinates $k \in \mathbb{R}^3$ and continuous latent variable $z$, the decoder outputs distribution parameters for a Gaussian distribution over $\hat{V}(k, z)$, i.e. the electron density of volume $\hat{V}_z$ at frequency $k$ in Fourier space. Unlike a standard deconvolutional decoder which produces a separate distribution for each voxel of a $D^3$ lattice given the latent variable, following spatial-VAE, we model a function over Cartesian coordinates (Bepler et al. (2019)). Here, these coordinates are explicitly treated as each pixel's location in 3D Fourier space and thus enforce the topological constraints between 2D views in 3D via the Fourier slice theorem.

By the image formation model, each image corresponds to an *oriented* central slice of the 3D volume in the Fourier domain (Section 2). During training, the 3D coordinates of an image's pixels can be explicitly represented by the rotation of a DxD lattice initially on the x-y plane. Under this model, the log probability of an image, $\hat{X}$, represented as a vector of size DxD, given the current MLP, latent pose variables $R \in SO(3)$ and $t \in \mathbb{R}^2$, and unconstrained latent variable, $z$, is:

$$\log p(\hat{X}|R, t, z) = \log p(\hat{X}'|R, z) = \sum_i \log p_\theta(\hat{V}|R^T c_0^{(i)}, z) \qquad (6)$$

where $i$ indexes over the coordinates of a fixed lattice $c_0$. Note that $\hat{X}' = S(-t)\hat{X}$ is the centered image, where $S$ is the phase shift operator corresponding to image translation in real space. We define $c_0$ as a vector of 3D coordinates of a fixed lattice spanning $[-0.5, 0.5]^2$ on the x-y plane to represent the unoriented coordinates of an image's pixels.

Instead of directly supplying $k$, a fixed positional encoding of $k$ is supplied to the decoder, consisting of sine and cosine waves of varying frequency:

$$pe^{(2i)}(k_j) = sin(k_j D\pi(2/D)^{2i/D}), \ i = 1, ..., D/2; k_j \in k \qquad (7)$$

$$pe^{(2i+1)}(k_j) = cos(k_j D\pi(2/D)^{2i/D}), \ i = 1, ..., D/2; k_j \in k \qquad (8)$$

Without loss of generality, we assume a length scale by our definition of $c_0$ which restricts the support of the volume to a sphere of radius 0.5. The wavelengths of the positional encoding thus follow a geometric series spanning the Fourier basis from wavelength 1 to the Nyquist limit $(2/D)$ of the image data. While this encoding empirically works well for noiseless data, we obtain better results with a slightly modified featurization for noisy datasets consisting of a geometric series which excludes the top 10 percentile of highest frequency components of the noiseless positional encoding.

### 3.2 INFERENCE

We employ a standard VAE for approximate inference of the latent variable $z$, but use a global search to infer the pose $\phi = (R, t)$ using a branch and bound algorithm.

*Variational encoder:* As each cryo-EM image is a noisy projection of an instance of the volume at a random, unknown pose (viewing direction), the image encoder aims to learn a *pose-invariant* representation of the protein's structural heterogeneity. Following the standard VAE framework, the probabilistic encoder $q_\xi(z|\hat{X})$ is a MLP with variational parameters $\xi$ and Gaussian output with diagonal covariance. Given an input cryo-EM image $\hat{X}$, represented as a DxD vector, the encoder MLP outputs $\mu_{z|\hat{X}}$ and $\Sigma_{z|\hat{X}}$, statistics that parameterize an approximate posterior to the intractable true posterior $p(z|\hat{X})$. The prior on $z$ is a standard normal, $\mathcal{N}(0, \mathbf{I})$.

*Pose inference:* We perform a global search over $SO(3) \times \mathbb{R}^2$ for the maximum-likelihood pose for each image given the current decoder MLP and a sampled value of $z$ from the approximate posterior. Two techniques are used to improve the efficiency of the search over poses: (1) discretizing the search space on a uniform grid and sub-dividing grid points after pruning candidate poses with

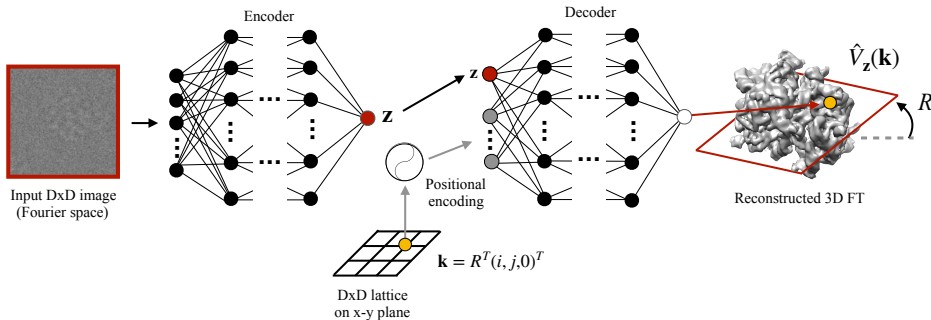

Figure 2: CryoDRGN model architecture. We use a VAE to perform approximate inference for latent variable $z$ denoting image heterogeneity. The decoder reconstructs an image pixel by pixel given $z$ and $pe(k)$, the positional encoding of 3D Cartesian coordinates. The 3D coordinates corresponding to each image pixel are obtained by rotating a DxD lattice on the x-y plane by $R$, the image orientation. The latent orientation for each image is inferred through a branch and bound global optimization procedure (not shown).

*branch and bound* (BNB), and (2) band pass limiting the objective to low frequency components and incrementally increasing the k-space limit at each iteration (*frequency marching*). The pose inference procedure encodes the intuition that low-frequency components dominate pose estimation, and is fully described in Appendix A.

In summary, for a given image $\hat{X}_i$, the image encoder produces $\mu_{z|\hat{X}_i}$ and $\Sigma_{z|\hat{X}_i}$. A sampled value of the latent $z_i \sim \mathcal{N}(\mu_{z|\hat{X}_i}, \Sigma_{z|\hat{X}_i})$ is broadcast to all pixels. Given $z_i$ and the current decoder, BNB orientational search identifies the maximum likelihood rotation $R_i$ and translation $t_i$ for $\hat{X}_i$. The decoder $p_\theta$ then reconstructs the image pixel by pixel given the positional encoding of $R_i^T c_0$ and $z_i$. The phase shift corresponding to $t_i$ and optionally the microscope CTF $\hat{g}_i$ is then applied on the reconstructed pixel intensities. Following the standard VAE framework, the optimization objective is the variational lower bound of the model evidence:

$$\mathcal{L}(\hat{X}_i; \xi, \theta) = \mathbb{E}_{q_\xi(z|\hat{X}_i)}[\log p_\theta(\hat{X}_i|z)] - KL(q_\xi(z|\hat{X}_i)||p(z)) \tag{9}$$

where the expectation of the log likelihood is estimated with one Monte Carlo sample. By comparing many 2D slices from the imaging dataset, the volume can be learned through feedback from these single views. Furthermore, this learning process is denoising as overfitting to noise from a single image would lead to higher reconstruction error for other views. We note that the distribution of 3D volumes models heterogeneity within a single imaging dataset, capturing structural variation for a particular protein or biomolecular complex, and that a separate network is trained per experimental dataset. Unless otherwise specified, the encoder and decoder networks are both MLPs containing 10 hidden layers of dimension 128 with ReLU activations. Further architecture and implementation details are given in Appendix A.

## 4 RELATED WORK

**Homogeneous cryo-EM reconstruction:** Cryo-EM reconstruction is typically accomplished in two stages: 1) generation of an initial low-resolution model followed by 2) iterative refinement of the initial model with a coordinate ascent procedure alternating between projection matching and refinement of the structure. In practice, initial structures can be obtained experimentally (Leschziner & Nogales (2006)), inferred based on homology to complexes with known structure, or via *ab-initio* reconstruction with stochastic gradient descent (Punjani et al. (2017)). Once an initial model is generated, there are many tools for iterative refinement of the model (Scheres (2012b); Punjani et al. (2017); Hohn et al. (2007); Lyumkis, Dmitry et al. (2013); Tang et al. (2007)). For example, Scheres (2012a) presents a Bayesian approach based on a probabilistic model of the image formation process and refines the structure via Expectation Maximization. Frequency marching is used extensively in existing tools to speed up the search for the optimal pose for each image (Scheres (2012b); Barnett et al. (2016); Punjani et al. (2017)). CryoSPARC implements a branch and bound optimization scheme, where their bound is a probabilistic lower bound based on the noise characteristics from the

image formation model (Punjani et al. (2017)). Ullrich et al. (2019) propose a differentiable voxel-based representation for the volume and introduce a variational inference algorithm for homogeneous reconstruction with known poses.

**Heterogeneous cryo-EM reconstruction:** In the cryo-EM literature, standard approaches for addressing structural heterogeneity use mixture models of discrete, independent volumes, termed *multiclass refinement* (Scheres (2010); Lyumkis, Dmitry et al. (2013)). These mixture models assume that the clusters are independent and homogeneous, and in practice require many rounds of expert-guided hierarchical clustering from appropriate initial volumes and manual choices for number of clusters. More recently, Nakane et al. (2018) extend the image generative model to model the protein as a sum of rigid bodies (determined from a homogeneous reconstruction), thus imposing structural assumptions on the type of heterogeneity. Frank & Ourmazd (2016) aim to build a continuous manifold of the images, however their approach requires pose supervision and final structures are obtained by clustering the images along the manifold and reconstructing with traditional tools. Recent theoretical work for continuous heterogeneous reconstruction includes expansion of discrete 3D volumes in a basis of Laplacian eigenvectors (Moscovich et al. (2019)) and a general framework for modelling hyper-volumes (Lederman et al. (2019)) e.g. as a tensor product of spatial and temporal basis functions (Lederman & Singer (2017)). To our knowledge, our work is the first to apply deep neural networks to cryo-EM reconstruction, and in doing so, is the first that can learn a continually heterogeneous volume from real cryo-EM data.

**Neural network 3D reconstruction in computer vision:** There is a large body of work in computer vision on 3D object reconstruction from 2D viewpoints. While these general approaches have elements in common with single particle cryo-EM reconstruction, the problem in the context of computer vision differs substantially in that 2D viewpoints are not projections and viewing directions are typically known. For example, Yan et al. (2016) propose a neural network that can predict a 3D volume from a single 2D viewpoint using only 2D image supervision. Gadelha et al. (2017) learn a generative model over 3D object shapes based on 2D images of the objects thereby disentangling variation in shape and pose. Tulsiani et al. (2018) also reconstruct and disentangle the shape and pose of 3D objects from 2D images by enforcing geometric consistency. These works attempt to encode the viewpoint 'projection' operation [2] explicitly in the model in a manner similar to our use of the Fourier slice theorem.

**Coordinate-based neural networks in computer vision:** Using spatial (i.e. pixel) coordinates as features to a convolutional decoder to improve generative modeling has been proposed many times, with recent work computing each image as a function of a fixed coordinate lattice and latent variables (Watters et al. (2019)). However, directly modeling a function that maps spatial coordinates to values is less extensively explored. In CocoNet, the authors present a deep neural network that maps 2D pixel coordinates to RBG color values. CocoNet learns an image model for single images, using the capacity of the network to memorize the image, which can then be used for various tasks such as denoising and upsampling (Bricman & Ionescu (2018)). Similarly, Spatial-VAE proposes a similar coordinate-based image model to enforce geometric consistency between rotated 2D images in order to learn latent image factors and disentangle positional information from image content (Bepler et al. (2019)). Our method extends many of these ideas from simpler 2D image modelling to enable 3D cryo-EM reconstruction in the Fourier domain.

## 5 RESULTS

Here, we present both qualitative and quantitative results for 1) homogeneous cryo-EM reconstruction, validating that cryoDRGN reconstructed volumes match those from existing tools; 2) heterogeneous cryo-EM reconstruction with pose supervision, demonstrating automatic learning of the latent manifold that previously required many expert-guided rounds of multiclass refinement; and 3) fully unsupervised reconstruction of continuous distributions of 3D protein structures, a capability not provided by any existing tool.

---

[2]This is not the meaning of projection in the context of this work, where it refers to *integration* along the imaging axis.

## 5.1 Unsupervised homogeneous reconstruction

We first evaluate cryoDRGN on homogeneous datasets, where existing tools are capable of reconstruction. We create two synthetic datasets following the cryo-EM image formation model (image size D=128, 50k projections, with and without noise), and use one real dataset from EMPIAR-10028 consisting of 105,247 images of the 80S ribosome downsampled to image size D=90. The encoder network is not used in homogeneous reconstruction. As a baseline for comparison, we perform homogeneous *ab-initio* reconstruction followed by iterative refinement in cryoSPARC (Punjani et al. (2017)). We compare against cryoSPARC as a representative of traditional state-of-the-art tools, which all implement variants of the same algorithm (Section 2). Further dataset preprocessing and training details are given in Appendix B.

We find that cryoDRGN inferred poses and reconstructed volumes match those from state of-the-art tools. The similarity of the volumes to the ground truth can be quantified with the with the Fourier shell correlation (FSC) curve[3]. Reconstructed volumes and quantitative comparison with the FSC curve is given in Figure S5. Pose error to the ground truth image poses are

| Method | Dataset | |
| --- | --- | --- |
| | No Noise | SNR=0.1 |
| cryoSPARC | 0.0009 / 0.47 | 0.002 / 0.64 |
| cryoDRGN | 0.0004 / 0.27 | 0.003 / 0.38 |

Table 1: Homogeneous reconstruction pose accuracy quantified by median rotation/translation error to the ground truth image poses. Rotation/translation error is defined as the Frobenius/L2 norm after alignment.

given in Table 1. For the real cryoEM dataset (no ground truth), the median pose difference between cryoDRGN and cryoSPARC reconstructions is 0.002 for rotations and 1.0 pixels for translations, and the resulting volumes are correlated above a FSC cutoff of 0.5 across all frequencies.

## 5.2 Heterogeneous reconstruction with pose supervision

Next, we evaluate cryoDRGN for heterogeneous cryo-EM reconstruction on EMPIAR-10076, a real dataset of the *E. coli* large ribosomal subunit (LSU) undergoing assembly (131,899 images, downsampled to D=90) (Davis et al. (2016)). Here, poses are obtained through alignment to an existing structure of the LSU and treated as known during training. In the original analysis of this dataset, multiple rounds of discrete multiclass refinement with varying number of classes followed by human comparison of similar volumes were used to identify 4 major structural states of the LSU. We train cryoDRGN with a 1-D latent variable treating image pose as fixed to skip BNB pose inference. As a baseline, we reproduce the published structures originally obtained through multiclass refinement with cryoSPARC. Further baseline and training details are given in Appendix C.

We find that CryoDRGN automatically identifies all 4 major states of the LSU (Figure 3a). Quantitative comparison with FSC curves[3] and additional volumes along the latent space are shown in Figure S7. We compare the cryoDRGN latent encoding $\mu_{z|X}$ for each image to the MAP cluster assignment in cryoSPARC and find that the learned latent manifold aligns with cryoSPARC clusters (Figure 3b). Cryo-DRGN identifies subpopulations in some of the cryoSPARC clusters (e.g. Class D), which is partitioned by a subsequent round of cryoSPARC multiclass refinement (Figure S8). Published structures A and F correspond to impurities in the sample. CryoDRGN correctly assigns images from these impurities to distinct clusters, but does not learn their correct structure since the poses inferred from aligning to the LSU template structure are incorrect.

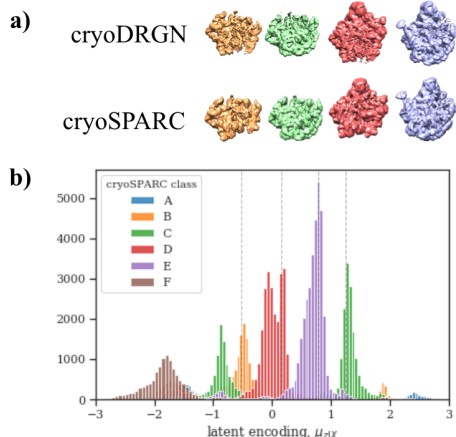

Figure 3: a) Volumes generated at values of the latent (at dashed lines) match the published volumes of the 4 major states B-E of the LSU. b) Distribution of images in the latent space, colored by cluster assignment from a discrete multiclass reconstruction in cryoSPARC.

[3]The FSC curve measures correlation between volumes as a function of radial shells in Fourier space. The field currently lacks a rigorous method for measuring the quality of reconstruction. In practice, however, resolution is often reported as $1/k_0$ where $k_0 = \arg\max_k FSC(k) < C$ and $C$ is some fixed threshold.

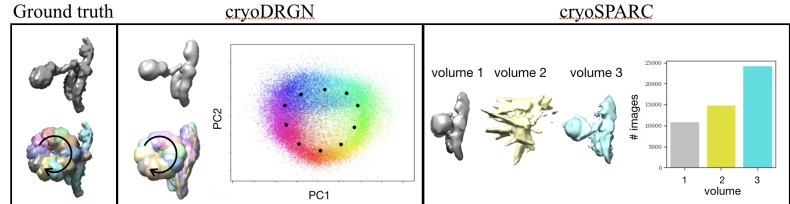

Figure 4: *Left:* Ground truth volume containing a continuous circular 1D motion. *Middle:* Reconstructed structures from cryoDRGN match the ground truth volumes with the correct continuous deformation. We visualize 10 structures (superimposed) sampled at the depicted points in the latent space. The distribution of images in the latent space (visualized in 2D with PCA) matches the topology of the true data manifold. *Right:* Reconstructed volumes from discrete 3-class reconstruction in cryoSPARC and the distribution of images over the three reconstructed volumes.

| Dataset | cryoDRGN | cryoDRGN+tilt | cryoSPARC |
|---|---|---|---|
| Linear 1D motion | 2.50(0.62) | 2.35(0.36) | 3.60(2.27) |
| Linear 2D motion | 4.44(2.50) | 2.93(1.02) | 6.90(3.77) |
| Circular 1D motion | 4.05(2.40) | 2.63(0.74) | 4.87(2.17) |
| Discrete 10 class | 4.95(3.16) | 2.58(1.00) | 5.69(5.15) |

Table 2: Reconstruction accuracy quantified by an FSC=0.5 resolution metric between the reconstructed volumes corresponding to each image and its ground truth volume. We report the average and standard deviation across 100 images in the dataset (lower is better; best possible is 2 pixels).

## 5.3 UNSUPERVISED HETEROGENEOUS RECONSTRUCTION

We test the ability of cryoDRGN to perform fully unsupervised heterogeneous reconstruction from datasets with different latent structure. We generate four datasets (each 50k projections, D=64) from an atomic model of a protein complex, containing either a 1D continuous motion, 2D continuous motion, 1D continuous circular motion, or a mixture of 10 discrete conformations (Figure S7). We train cryoDRGN with a 1D latent variable for the linear 1D dataset and a 10D latent variable for the other 3 datasets. As a baseline, we perform multiclass reconstruction in cryoSPARC sweeping K=2-5 classes. We compare against K=3, which had the best qualitative results.

We also propose a modification to cryoDRGN in order to train on *tilt series pairs* datasets. Tilt series pairs is a variant of cryo-EM in which, for each image $X_i$, a corresponding image $X_i'$ is acquired after tilting the imaging stage by a known angle. This technique was originally employed to identify the chirality of molecules (Belnap et al. (1997)), which is lost in the projection from 3D to 2D. We propose using tilt series pairs to encourage invariance of $q_\xi$ with respect to pose transformations for a given $\hat{V}_\mathbf{z}$ (and incidentally to identify the chirality of $\hat{V}_\mathbf{z}$). We make minor modifications to the architecture as described in Appendix D.

In Figure 4, we show that cryoDRGN reconstructed volumes for the circular 1D dataset qualitatively match the ground truth structures. Note that while we only visualize 10 structures sampled along the latent space, the volume decoder can reconstruct the full continuum of states. In contrast, cryoSPARC multiclass reconstruction, a discrete mixture model of independent structures, is only able to reconstruct 2 (originally unaligned) structures which resemble the ground truth. Volumes contain blurring artifacts from clustering images from different conformations into the assumed-homogeneous clusters in the mixture model. Results for the remaining datasets are given in Figures S10-13.

We quantitatively measure performance on this task with an FSC resolution metric computed between the MAP volume for each image $V_{z_i|\hat{X}_i}$ and the ground truth volume which generated each image, averaged across images in the dataset (Table S4). We find that cryoDRGN reconstruction accuracy is much higher than state-of-the-art discrete multiclass reconstruction in cryoSPARC, with further improvement achieved by training on tilt series pairs.

## 6    CONCLUSIONS

We present a novel neural network-based reconstruction method for single particle cryo-EM that learns continuous variation in protein structure. We applied cryoDRGN on a real dataset of highly heterogeneous ribosome assembly intermediates and demonstrate automatic partitioning of structural states. In the presence of simulated continuous heterogeneity, we show that cryoDRGN learns a continuous representation of structure along the true reaction coordinate, effectively disentangling imaging orientation from intrinsic structural heterogeneity. The techniques described here may also have broader applicability to image and volume generative modelling in other domains of computer vision and 3D shape reconstruction.

### ACKNOWLEDGMENTS

We thank Ben Demeo, Ashwin Narayan, Adam Lerer, Roy Lederman, and Kotaro Kelley for helpful discussions and feedback. This work was funded by the National Science Foundation Graduate Research Fellowship Program, NIH grant R01-GM081871, NIH grant R00-AG050749, and the MIT J-Clinic for Machine Learning and Health.

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

# A APPENDIX - METHODS

## A.1 BRANCH AND BOUND IMPLEMENTATION DETAILS

We perform a global search over $SO(3) \times \mathbb{R}^2$ for the maximum-likelihood pose for each image given the current decoder MLP. Two techniques are used to improve the efficiency of the search over poses: (1) discretizing the search space on a uniform grid and sub-dividing grid points after pruning candidate poses with *branch and bound*, and (2) band pass limiting the objective to low frequency components and incrementally increasing the k-space limit at each iteration (*frequency marching*).

Our branch and bound algorithm for pose optimization is given in Algorithm 1. Briefly, we discretize $SO(3)$ uniformly using the Hopf fibration Yershova et al. (2010) at a predefined base resolution of the grid and incrementally increase the grid resolution by sub-dividing grid points. At each resolution of the grid, the set of candidate poses is pruned using a branch and bound (BNB) optimization scheme, which alternates between a computationally inexpensive lower bound on the objective function evaluated at all grid points and an upper bound consisting of the true objective evaluated on the best lower-bound candidate. Grid points whose lower bound is higher than this value are excluded for subsequent iterations. In our case, the loss is evaluated on low-frequency components of the image; specifically, Fourier components with $|\mathbf{k}| < k_{max}$ is an effective lower bound, as it is both inexpensive to compute and captures most of the power (and thus the error). This bound encodes the intuition that low-frequency components dominate pose estimation. We concomitantly increase $k_{max}$ at each iteration of grid subdivision.

At each iteration, some poses are excluded by BNB, and the remaining poses are further discretized. Although BNB is risk-free in the sense that the optimal pose at a given resolution will not be pruned, our application of it is not risk-free as a candidate pose with high loss at a given resolution doesn't guarantee that its neighbor in the next iteration will not have a lower loss. Irrespective, in practice, we find that at a sufficiently fine base resolution, we obtain good results on a tractable timescale (hours on a single GPU).[4]

We reimplement the uniform multiresolution grids on $SO(3)$ based on Yershova et al. (2010), using the Healpix Gorski et al. (2005) grid for the sphere and the Hopf fibration to uniformly lift the grid to $SO(3)$. The base grid on $SO(3)$ contains 576 orientations. We use the ordinary grid for translations containing $7^2$ points with an extent of 20 pixels for D=128 datasets. We subdivide the grid 5 times for a final resolution of 0.92 degrees for the orientation and 0.08 pixels for the translation. For D=64 datasets, we use a translational grid with extent of 10 pixels.

---

**Algorithm 1** CryoDRGN branch and bound with frequency marching

---

1: **procedure** OPTPHI($\hat{X}, \hat{V}_{\mathbf{z}}$)          ▷ Find the optimal image pose given the current decoder
2:     $k_{min} \leftarrow 12,\ k_{max} \leftarrow D/2,\ N_{iter} \leftarrow 5$
3:     $\Phi \leftarrow SO(3) \times \mathbb{R}^2$ grid at base resolution
4:     $k \leftarrow k_{min}$
5:     **for** $iter = 1 \ldots N_{iter}$ **do**
6:         **for** $\phi_i \in \Phi$ **do**                    ▷ Compute lower bound at all grid points
7:             $lb(\phi_i) \leftarrow$ loss between $\hat{X}$ and SLICE($\hat{V}_{\mathbf{z}}, \phi_i$) at $\mathbf{k} < k$
8:         $\phi^* \leftarrow \arg\min(lb)$
9:         $ub \leftarrow$ loss between $\hat{X}$ and SLICE($\hat{V}_{\mathbf{z}}, \phi^*$) at $\mathbf{k} < k_{max}$          ▷ Compute upper bound
10:         $\Phi_{new} \leftarrow \{\}$
11:         **for** $\phi_i \in \Phi$ **do**                    ▷ Subdivide grid points below the upper bound
12:             **if** $lb(\phi_i) < ub$ **then**
13:                 $\Phi_{new} \leftarrow \Phi_{new} \cup$ SUBDIVIDE($\phi_i$)
14:         $\Phi \leftarrow \Phi_{new}$
15:         $k \leftarrow k + (k_{max} - k_{min})/(N_{iter} - 1)$          ▷ Increase frequency band limit
16:     **return** $\phi^*$

---

[4]The difference in loss between nearby poses could be incorporated into the BNB lower bound, but this would require assumptions about the smoothness of the loss with respect to pose. We leave this detail for future work.

## A.2 TRAINING DETAILS

Given an imaging dataset, $\hat{X}_1, ... \hat{X}_N$, we summarize three training paradigms of cryoDRGN. 1) For homogeneous reconstruction, we only train the volume decoder $p_\theta$ and perform BNB pose inference for the unknown $\phi_i$'s for each image. 2) As an intermediate task, we can perform heterogeneous reconstruction training the image encoder $q_\xi$ and the volume decoder $p_\theta$ with known $\phi_i$'s to skip BNB pose inference. 3) For fully unsupervised heterogeneous reconstruction, we jointly train $q_\xi$ and $p_\theta$ to learn a continuous latent representation, performing BNB pose inference for the unknown pose of each image.

Unless otherwise specified, the encoder and decoder networks are both MLPs containing 10 hidden layers of dimension 128 with ReLU activations. A fully connected architecture is used instead of a convolutional architecture because the images are not represented in real space.

Instead of representing both the real and imaginary components of each image, we use the closely-related Hartley space representation (Hartley (1942)). The Hartley transform of real-valued functions is equivalent to the real minus imaginary component of the FT, and thus is real valued. The Fourier slice theorem still holds and the error model is equivalent.

In this work, we simplify the image generation model to Gaussian white noise. Therefore, for a given image, the negative log likelihood for a reconstructed slice from the decoder corresponds to the mean squared error between the phase-shifted image and the oriented slice from the volume decoder. We leave the implementation of a colored noise model to future work.

We use the Adam optimizer (Kingma & Ba (2014)) with learning rate of 5e-4 for experiments involving noiseless, homogeneous datasets, and 1e-4 for all other experiments. All models are implemented in Pytorch (Paszke et al. (2017)).

## B HOMOGENEOUS RECONSTRUCTION

### B.1 DATASET PREPARATION

*Simulated datasets:* From a ground truth 3D volume, we simulated datasets following the cryo-EM image formation model by 1) rotating the 3D volume in real space by $R$, where $R \in SO(3)$ is sampled uniformly, 2) projecting (integrating) the volume along the z-axis, 3) shifting the resulting 2D image by $t$, where $t$ is sampled uniformly from $[-10, 10]^2$ pixels, and 4) optionally adding noise to an SNR of 0.1, a typical value for cryo-EM data (Baxter et al. (2009)). Following convention in the cryo-EM field, we define SNR as the ratio of the variance of the signal to the variance of the noise. We define the noise-free signal images to be the entire DxD image. 50k projections were generated for each dataset with image size of D=128.

*Real dataset:* To generate the real cryo-EM dataset for homogeneous reconstruction, images from EMPIAR-10028 (Wong et al. (2014)) were downsampled by a factor of 4 by clipping in Fourier space. The images were then 'phase flipped' in Fourier space by their contrast transfer function, a given real-valued function with range [-1,1] determined by the microscopy conditions, i.e. the Fourier components are negated where the CTF is negative.

### B.2 TRAINING

For each dataset, we train the volume decoder (10 hidden layers of dimension 128) in minibatches of 10 images with random orientations for the first epoch to learn a volume with roughly correct spatial extent, followed by 4 epochs with branch and bound (BNB) pose inference (30 min/epoch noiseless, 80 min/noisy datasets). Since BNB pose inference is the bottleneck during training, we employ a multiscale training protocol, where after 4 epochs with BNB pose inference, the latent pose is fixed, and we train a separate, larger volume decoder (10 hidden layers of dimension 500) for 15 epochs with fixed poses to "refine" the structure to high resolution (20 min/epoch). Training times are reported for 50k, D=128 image datasets trained on a Nvidia Titan V GPU.

## B.3 SUPPLEMENTARY RESULTS

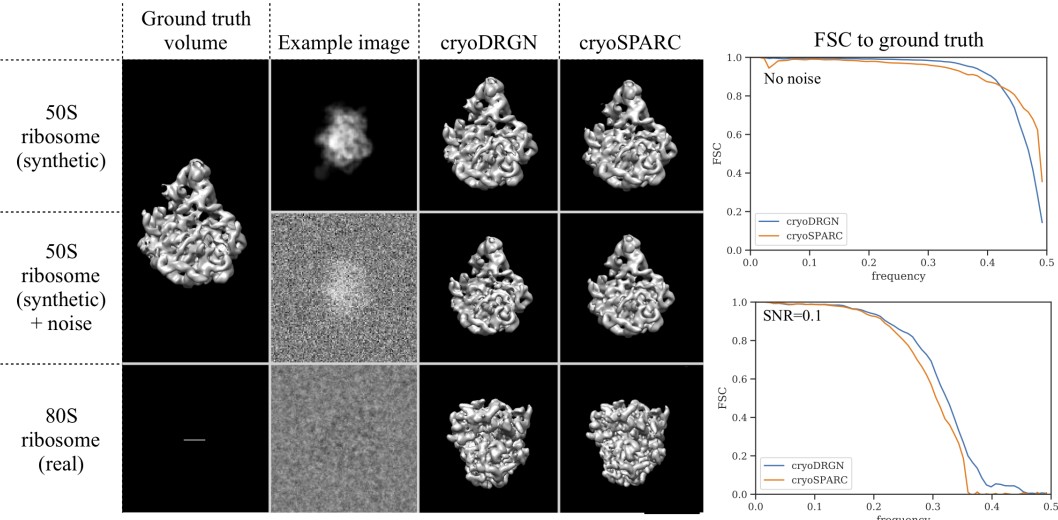

Figure S5: *Left:* CryoDRGN unsupervised homogeneous reconstruction on 2 simulated datasets and 1 real cryo-EM dataset matches state-of-the-art. *Right:* Fourier shell correlation (FSC) curves between the reconstructed volume and the ground truth volume for the synthetic ribosome datasets.

# C  HETEROGENEOUS RIBOSOME RECONSTRUCTION WITH POSE SUPERVISION

*Dataset preparation:* We used the dataset from EMPIAR-10076 which contains 131,899 images of the *E. coli* large ribosomal subunit (LSU) in various stages of assembly (Davis et al. (2016)). Images were downsampled to D=128 by clipping in Fourier space. Poses were determined by aligning the images to a mature LSU structure obtained from a homogeneous reconstruction of the full resolution dataset in cryoSPARC, i.e. "a consensus reconstruction".

*Baseline:* In the original analysis of this dataset, multiple rounds of multiclass refinement in sweeps of varying number of classes followed by expert manual alignment and clustering of similar volumes were used to identify 6 classes, labeled A-F consisting of 4 major structural states of the LSU (classes B-E) and 2 additional structures of the 70S and 30S ribosome, class A and F, respectively.

Since the published dataset did not contain the corresponding image cluster assignments, we perform multiclass refinement in cryoSPARC using the published structures of the 6 major states, low pass filtered to 25Å as initial models, to reproduce the results and obtain image cluster assignments. Aside from class A and F (low population impurities in the sample), the remaining structures correlate well with the published volumes (Figure S6).

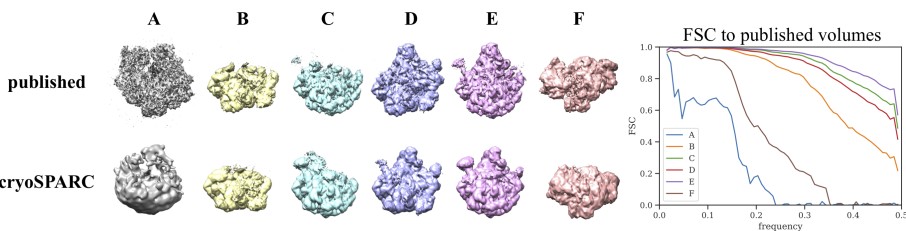

Figure S6: Reconstructed volumes from cryoSPARC multiclass refinement using the published structures of the 6 major states, low pass filtered to 25Åas initial models. Right: FSC curves between the cryoSPARC reconstructed and published volumes.

*cryoDRGN training:* We train cryoDRGN with a 1-D latent variable in minibatches of 10 images for 200 epochs, treating image pose as fixed (11 min/epoch on a Nvidia Titan V GPU). To simplify representation learning for $q_\xi$, we center and phase flip images before inputting to the encoder. We encode and decode a circle of pixels with diameter D=128 instead of the full 128x128 image.

## C.1 Supplementary results

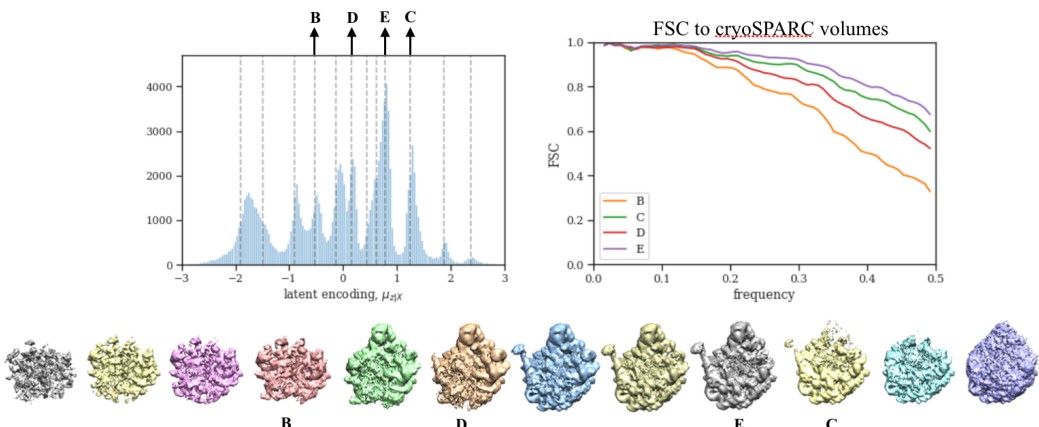

Figure S7: *Left:* Latent encoding for each image of the dataset from EMPIAR-10076. *Bottom:* Volumes from 12 sampled values along the latent space (dashed lines). *Right:* Fourier shell correlation (FSC) curves for 4 structures against the published volumes for classes B-E from corresponding to structural states of the large ribosomal subunit during assembly (Davis et al. (2016)).

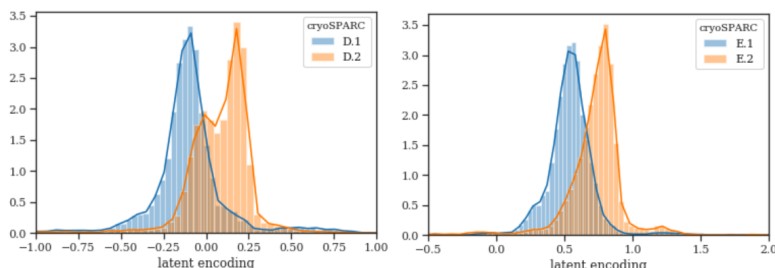

Figure S8: The latent encoding aligns with cluster assignments from a successive round of multiclass refinement in cryoSPARC on the subset of images from class D and E.

## D Fully unsupervised heterogeneous reconstruction

### D.1 Dataset preparation

*Linear 1D motion:* We generated a dataset containing one continuous degree of freedom as follows: From an atomic model of a protein complex, a single bond in the atomic model was rotated while keeping the remaining structure fixed, and 50 atomic models were sampled along this reaction coordinate. 1000 projections with random rotations and in-plane translations were generated for each model, yielding a total of 50k images, approximating a uniform distribution along a continuous reaction coordinate.

*Linear 2D motion:* We extended the linear 1D motion dataset by introducing a second degree of freedom from rotating a bond in the atomic model that connected a different protein in the complex. Similar to the 1D motion dataset, from a starting configuration, the original bond was rotated +/- N degrees, and 50 models were sampled along this reaction coordinate. Then from the starting

conformation, the second bond was rotated +/- 90 degrees, and 50 additional models were sampled along the second reaction coordination. 500 projections were generated from each model, yielding a total of 50k images.

*Circular 1D motion:* For this dataset, we rotated a bond a full 360 degrees and sample 100 models along this circular reaction coordinate. 500 projections were generated from each model, yielding a total of 50k images.

*Discrete 10 class:* For this dataset, we sampled 10 random configurations for the proteins in the complex. 5000 projection images were generated from each model, yielding a dataset containing a mixture of 10 discrete states.

For all four datasets, random rotations were generated uniformly from $SO(3)$, and translations were sampled uniformly from $[-5, 5]$ pixels. The image size was D=64 with absolute spatial extent of 720Åand Nyquist limit of 22.5Å. Schematics of the simulated motions are given in Figure S9.

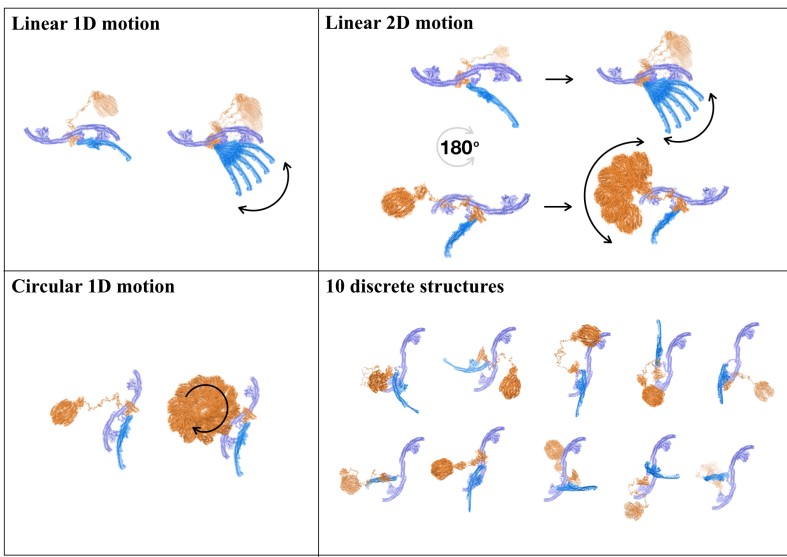

Figure S9: Ground truth atomic model and the heterogeneity introduced for different datasets.

## D.2 Tilt series pairs

Tilt series pairs is a variant of cryo-EM in which, for each image $X_i$, a corresponding image $X_i'$ is acquired after tilting the imaging stage by a known angle. This technique was originally employed to identify the chirality of molecules (Belnap et al. (1997)), which is lost in the projection from 3D to 2D and therefore cannot be inferred from standard cryo-EM. Inferential procedures such as expectation maximization converge to one handedness or the other depending on their initialization. In multiclass reconstruction, different classes are not guaranteed to possess the same handedness even if there is a high relatedness between structures. We remark on this experimental technique as we propose using tilt series pairs to encourage invariance of $q_\xi$ with respect to pose transformations for a given $\hat{V}_z$ (and incidentally also to identify the chirality of $\hat{V}_z$). To train on tilt series pairs, the encoder is split into two MLPs, the first learning an intermediate encoding of each image, and the second mapping the concatenation of the two encodings to the latent space. We use an 8 layer MLP with output dimension 128 for the former and a 2 layer MLP with input dimension 256 for the latter. All hidden layers have dimension 128. For branch and bound, the combined loss over both images is evaluated for each grid point of $SO(3) \times \mathbb{R}^2$. To generate the image $X_{tilt,i}$ associated with $X_i$, prior to rotating the volume by $R_i$, we rotate the volume by a constant 45 degrees around the x-axis.

### D.3 TRAINING

We trained cryoDRGN in minibatches of 5 images for 40 epochs without tilt series pairs and 20 epochs with tilt series pairs. We trained a 1-D latent variable for the linear 1D motion dataset, and 10-D latent variables for the remaining datasets. Random angles were used for the first epoch of training to learn roughly the correct spatial extent of the volume and BNB pose inference was used for the remaining epochs. The runtime was 120 min/epoch vs 2 min/epoch with and without BNB pose inference, respectively, on a Nvidia Titan V GPU.

### D.4 SUPPLEMENTARY RESULTS

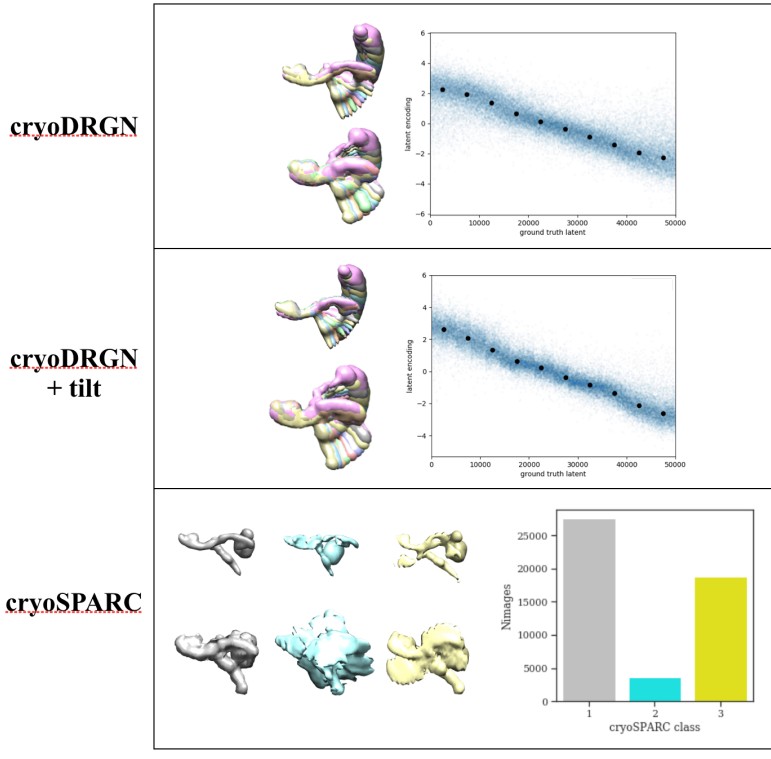

Figure S10: Reconstruction results for the linear 1D dataset by cryoDRGN and by discrete multiclass reconstruction in cryoSPARC. *Top:* Reconstructed structures from cryoDRGN sampled along the latent space (at depicted points) matches the ground truth variation. The predicted latent encoding correlates with the ground truth latent degree of freedom. *Middle:* CryoDRGN results with tilt series *Bottom:* Reconstructed volumes and the distribution of images over clusters from discrete multiclass reconstruction in cryoSPARC. Volumes are visualized at high and low isosurface, showing artifacts in the cryoSPARC structures.

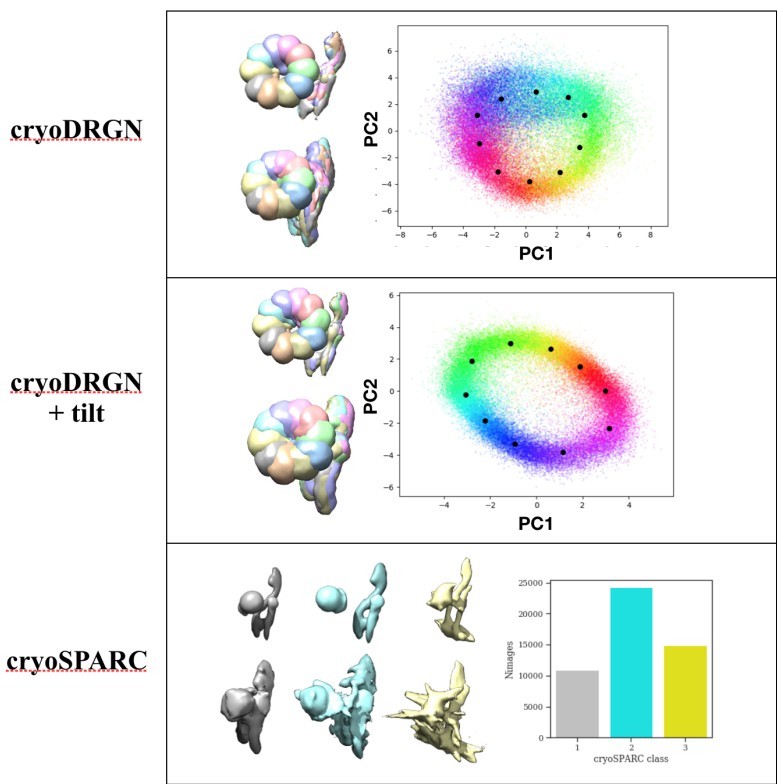

Figure S11: Reconstruction results for the circular 1D dataset by cryoDRGN and by discrete multiclass reconstruction in cryoSPARC. *Top:* Reconstructed structures from cryoDRGN sampled along the latent space (at depicted points) matches the ground truth variation. The distribution of images in the latent space matches the ciruclar topology of the true data manifold. *Middle:* CryoDRGN results with tilt series *Bottom:* Reconstructed volumes and the distribution of images over clusters from discrete multiclass reconstruction in cryoSPARC. Volumes are visualized at high and low isosurface, showing artifacts in the cryoSPARC structures.

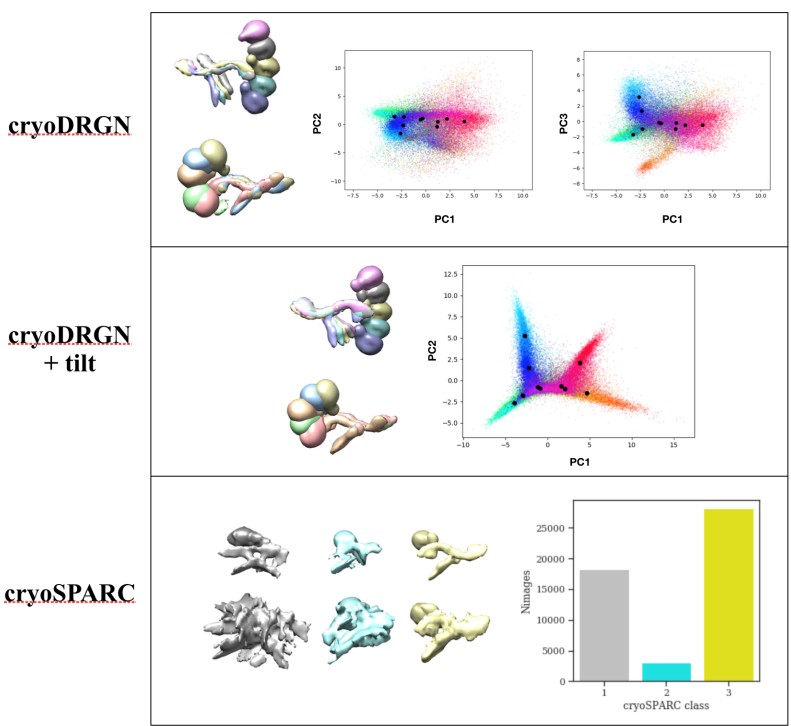

Figure S12: Reconstruction results for the linear 2D dataset by cryoDRGN and by discrete multiclass reconstruction in cryoSPARC. *Top:* Reconstructed structures from cryoDRGN sampled along the latent space (at depicted points) roughly matches the ground truth variation, however the distribution of images in the latent space does not recapitulate the true data manifold well. *Middle:* CryoDRGN results with tilt series reconstructs the true structural variation and the distribution of images in the latent space matches the topology of the true data manifold. *Bottom:* Reconstructed volumes and the distribution of images over clusters from discrete multiclass reconstruction in cryoSPARC. CryoSPARC volumes are visualized at high and low isosurface, showing artifacts at low isosurface

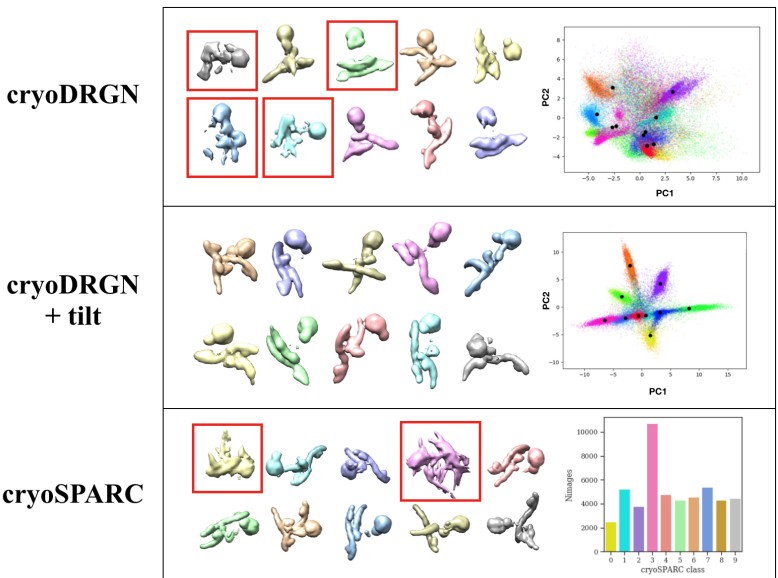

Figure S13: Reconstruction results for the dataset containing 10 discrete structures by cryoDRGN and by discrete multiclass reconstruction in cryoSPARC. *Top:* The majority of reconstructed structures from cryoDRGN sampled along the latent space (at depicted points) matches the ground truth structures, however some are incorrect (red boxes), and the learned data manifold is not well separated into clusters. *Middle:* CryoDRGN results with tilt series reconstructs the 10 structures and clusters the images in the latent space accordingly. *Bottom:* Reconstructed volumes from discrete multiclass reconstruction in cryoSPARC and the distribution of images over clusters. CryoSPARC learns 8 out of 10 structures correctly.

|  | cryoSPARC | | | |
| Dataset | K=2 | K=3 | K=4 | K=5 |
| --- | --- | --- | --- | --- |
| Linear 1D motion | 5.11(3.82) | 3.60(2.27) | 7.40(4.16) | 7.59(4.58) |
| Linear 2D motion | 6.89(2.21) | 6.90(3.77) | 5.98(2.10) | 6.76(4.47) |
| Circular 1D motion | 5.16(2.70) | 4.87(2.17) | 7.50(3.32) | 4.62(1.93) |

Table S3: Relationship between number of classes in cryoSPARC and reconstruction accuracy quantified by an FSC=0.5 resolution metric between the reconstructed volumes corresponding to each image and its ground truth volume. We report the average and standard deviation across 100 images in the dataset (lower is better; best possible is 2 pixels).

|  | cryoDRGN | | | cryoDRGN+tilt | | |
| Dataset | z-D=1 | z-D=2 | z-D=10 | z-D=1 | z-D=2 | z-D=10 |
| --- | --- | --- | --- | --- | --- | --- |
| Linear 1D motion | 2.50(0.62) | 2.34(0.12) | – | 2.35(0.36) | 2.43(0.26) | – |
| Linear 2D motion | 7.16(4.69) | 4.38(3.15) | 4.44(2.50) | 3.38(1.18) | 2.97(1.24) | 2.93(1.02) |
| Circular 1D motion | 5.61(4.36) | 4.95(2.91) | 4.05(2.40) | 3.12(0.96) | 2.65(0.67) | 2.63(0.74) |

Table S4: Relationship between $z$ dimension in cryoDRGN and reconstruction accuracy quantified by an FSC=0.5 resolution metric between the reconstructed volumes corresponding to each image and its ground truth volume. We report the average and standard deviation across 100 images in the dataset (lower is better; best possible is 2 pixels).

