# OpenReview forum: "Reconstructing continuous distributions of 3D protein structure from cryo-EM images"
_ICLR.cc/2020/Conference — Accept (Spotlight)_

### Official Review · AnonReviewer2 · 2019-10-23
**Official Blind Review #2**

**Rating:** 8

**Review:**

The authors introduce cryoDRGN, a VAE neural network architecture to reconstruct 3D protein structure from 2D cryo-EM images.

The paper offers for a good read and diagrams are informative.

Below are comments for improvement and clarification.

> Consider explaining cryoSPARC in detail given that is the state-of-the-art technique and to which all the cryoDGRN results are compared.

> In Figure 4 and the related experiment,  how are a) the cryoSPARK volumes related to cryoDRGN volumes, b) what do the clusters mean in cryoSPARK and how do they compare with the corresponding outputs of cryoDRGN

> What would runtime comparisons be for cryoSPARK and cryoDGRN, for an unsupervised heteregeneous reconstruction?

**Experience Assessment:**

I have read many papers in this area.

**Review Assessment: Checking Correctness Of Derivations And Theory:**

I assessed the sensibility of the derivations and theory.

**Review Assessment: Checking Correctness Of Experiments:**

I assessed the sensibility of the experiments.

**Review Assessment: Thoroughness In Paper Reading:**

I read the paper at least twice and used my best judgement in assessing the paper.

---

> ### Author Response · Authors · 2019-11-15
> **Response to reviewer #2**
>
> Thank you for your comments and questions. Classical cryo-EM reconstruction algorithms (e.g. cryoSPARC) are described in Section 2.2 at a high level and we refer the reader to its reference (Punjani et al. 2017) for more details on their implementation.
>
> To clarify the relationship between the cryoSPARC and cryoDRGN heterogeneous reconstruction in Figure 4, CryoSPARC imposes a discrete model for heterogeneity, specifically a mixture model of K volumes. The cryoSPARC results in Figure 4 are the volumes and the distribution of images over the 3 clusters from their unsupervised reconstruction. In contrast, the continuous latent variable from cryoDRGN unsupervised reconstruction is able to reconstruct the continuous motion of the ground truth volume. We have clarified the text to reduce any confusion and added training times for these methods to the appendix. Thank you for the recommendations!

---

### Official Review · AnonReviewer3 · 2019-10-23
**Official Blind Review #3**

**Rating:** 8

**Review:**

~The authors build a new method to recapitulate the 3D structure of a biomolecule from cryo-EM images that allows for flexibility in the reconstructed volume.~

I thought this paper is very well written and tackles a difficult project.

There is a previous work that these authors should cite:

Ullrich, K., Berg, R.V.D., Brubaker, M., Fleet, D. and Welling, M., 2019. Differentiable probabilistic models of scientific imaging with the Fourier slice theorem. arXiv preprint arXiv:1906.07582.

How does your method compare to this paper? In Ullrich et al., they report “Time until convergence, MSE [10^-3/voxel], and Resolution [Angstrom]). I think these statistics would be useful to report in your work, as they are more familiar with folks in the cryoEM field.

In Equation 3, how does one calculate Z, the normalization constant?

For the decoder, how large of the 3D space are you generating? What are the units? Are you using voxels to represent atomic density? What is the voxel size? Is it the same as on Page 11?

I think more description of the neural network architecture would be useful (more than what is reported on page 12).


**Experience Assessment:**

I have published one or two papers in this area.

**Review Assessment: Checking Correctness Of Derivations And Theory:**

I assessed the sensibility of the derivations and theory.

**Review Assessment: Checking Correctness Of Experiments:**

I carefully checked the experiments.

**Review Assessment: Thoroughness In Paper Reading:**

I read the paper thoroughly.

---

> ### Author Response · Authors · 2019-11-15
> **Response to reviewer #3**
>
> 1. Thank you for your comments and thank you in particular for pointing us to a reference we missed, which we have added to the manuscript.
>
> Ullrich et al. introduce some of the same foundational building blocks for applying differentiable models to the cryoEM reconstruction task. In particular, they propose a differentiable voxel-based representation for the volume and introduce a variational inference algorithm for learning the volume through gradient-based optimization.
>
> Due to their voxel-based representation, they introduce a method to differentiate through the 2D projection operator. In contrast, we parametrically learn a continuous function for volume via a coordinate-based MLP, which seamlessly allows differentiation through the slicing and rotation operators without having to deal with discretization.
>
> Their method is able to learn a homogeneous volume with given poses, whereas we perform fully unsupervised reconstruction of heterogeneous volumes. They show empirical experiments that highlight many of the challenges for variational inference of these models. In particular, inference of the unknown pose is challenging with gradient-based optimization and contains many local minima (their Fig 6), which we address with a branch and bound algorithm.
>
> We report a Fourier Shell Correlation (FSC) metric, which is a commonly used resolution metric in the cryoEM field. Voxel-wise MSE is not typically used in the cryoEM literature as it is sensitive to background subtraction and data normalization. We have added training times for these methods to the SI.
>
> 2. The normalization constant in Eq. 3 is the partition function over all possible values of the latent pose and volume. Instead of computing this (intractable) constant, coordinate ascent on the dataset log likelihood is used to refine estimates of pose and volume in traditional algorithms.
>
> 3. The extent of the 3D space is determined by the dataset’s image size and resolution. We define a lengthscale such that image coordinates are modeled on a fixed lattice spanning [-0.5, 0.5]^2 with grid resolution determined by the image size. The absolute spatial extent is thus determined by the Angstrom/pixel ratio for each dataset. Similarly, final volumes for a given value of the latent are generated by evaluating a 3D lattice with extent [-0.5,0.5]^3 with grid resolution determined by the dataset image size. We have added the absolute spatial extent to the description of each dataset in the revised manuscript.
>
> 4. We have included additional architectural details in the revised manuscript, and we will be releasing the source code which will hopefully further clarify the architecture.

---

> > ### Public Comment · ~Jaejun_Yoo1 · 2020-02-19
> > **About the source code release**
> >
> > Could you give us an approximate date of the code release?

---

> > > ### Author Response · Authors · 2020-02-21
> > > **Software release**
> > >
> > > Thanks for inquiring. We are working on a software package for non-expert users, and hope to release as soon as possible. Feel free to contact us directly if you’re interested in early access to the codebase.

---

### Official Review · AnonReviewer1 · 2019-10-24
**Official Blind Review #1**

**Rating:** 6

**Review:**

- The authors proposed a novel method for cryo-EM reconstruction that extends naturally to modeling continuous generative factors of structural heterogeneity. To address intrinsic protein structural heterogeneity, they explicitly model the imaging operation to disentangle the orientation of the molecule by formulating decoder as a function of Cartesian coordinates.

- The problem and the approach are well motivated.

- This reviewer has the following comments:
1) VAE is known to generate blurred images. Thus, based on this approach, the reconstruction image may not be optimal with respect to the resolution which might be critical for cryo-EM reconstruction. What's your opinion?
2) What's the relationship between reconstructed performance, heterogeneity of the sample and dimensions of latent space?
3) It would be interesting to show any relationship, reconstruction error with respect to the number of discrete multiclass.
4) How is the proposed method generalizable?

**Experience Assessment:**

I do not know much about this area.

**Review Assessment: Checking Correctness Of Derivations And Theory:**

I did not assess the derivations or theory.

**Review Assessment: Checking Correctness Of Experiments:**

I assessed the sensibility of the experiments.

**Review Assessment: Thoroughness In Paper Reading:**

I read the paper at least twice and used my best judgement in assessing the paper.

---

> ### Author Response · Authors · 2019-11-15
> **Response to reviewer #1**
>
> Thank you for your comments and questions. We have updated the manuscript to clarify these questions.
> 1) The VAE is hypothesized to produce blurry images when the inference/generative models are not sufficiently expressive for the data modeling task, and in particular due to the typical choice of MSE loss (i.e. Gaussian error model), thus blurring sharp edges in complex natural image data [1,2,3]. In the case of cryo-EM, the high noise in the images is typically assumed to be Gaussian and therefore using the MSE loss has a denoising effect. In our experiments, we were able to achieve resolutions up to the ground truth resolution or matching published structures with our architecture and training settings, though we agree with the reviewer that exploring alternative generative models is a promising future direction.
>
> [1] https://arxiv.org/abs/1611.02731
> [2]  https://openreview.net/pdf?id=B1ElR4cgg
> [3] https://arxiv.org/pdf/1702.08658.pdf
>
> 2) We observed accurate reconstructions as long as the dimension exceeded the dimension of the underlying data manifold and faster training with higher dimensional latent variables. We have added these results to the appendix in the revised manuscript.
> 3) We varied the number of classes for comparison against SOTA discrete multiclass reconstruction and selected 3 classes which had the lowest error for our comparison in Table 2. We have added these results to the appendix in the revised manuscript.
> 4) Our coordinate-based neural network model for volumes provides a general framework for modeling extrinsic orientational changes in a differentiable manner. This work could be applied in other domains of scientific imaging such as reconstruction of tomograms or CT scans.

---

### Decision · Program_Chairs · 2019-12-19

**Decision:**

Accept (Spotlight)

**Comment:**

The paper introduces a generative approach to reconstruct 3D images for cryo-electron microscopy (cryo-EM).

All reviewers really liked the paper, appreciate the challenging problem tackled and the proposed solution.

Acceptance is therefore recommended.